# RT-LAMP-Based Molecular Diagnostic Set-Up for Rapid Hepatitis C Virus Testing

**DOI:** 10.3390/bios12050298

**Published:** 2022-05-05

**Authors:** Sandhya Sharma, Emmanuel Thomas, Massimo Caputi, Waseem Asghar

**Affiliations:** 1Department of Electrical Engineering and Computer Science, Florida Atlantic University, Boca Raton, FL 33431, USA; ssharma2013@fau.edu; 2Asghar-Lab: Micro and Nanotechnology in Medicine, College of Engineering and Computer Science, Boca Raton, FL 33431, USA; 3Department of Microbiology and Immunology and Sylvester Comprehensive Cancer Center, University of Miami School of Medicine, Miami, FL 33136, USA; ethomas1@med.miami.edu; 4Charles E. Schmidt College of Medicine, Florida Atlantic University, Boca Raton, FL 33431, USA; mcaputi@health.fau.edu; 5Department of Biological Sciences (Courtesy Appointment), Florida Atlantic University, Boca Raton, FL 33431, USA

**Keywords:** RT-LAMP, point-of-care diagnostics, microfluidics, molecular HCV test, colorimetric detection

## Abstract

Hepatitis C virus (HCV) infections occur in approximately 3% of the world population. The development of an enhanced and extensive-scale screening is required to accomplish the World Health Organization’s (WHO) goal of eliminating HCV as a public health problem by 2030. However, standard testing methods are time-consuming, expensive, and challenging to deploy in remote and underdeveloped areas. Therefore, a cost-effective, rapid, and accurate point-of-care (POC) diagnostic test is needed to properly manage the disease and reduce the economic burden caused by high case numbers. Herein, we present a fully automated reverse-transcription loop-mediated isothermal amplification (RT-LAMP)-based molecular diagnostic set-up for rapid HCV detection. The set-up consists of an automated disposable microfluidic chip, a small surface heater, and a reusable magnetic actuation platform. The microfluidic chip contains multiple chambers in which the plasma sample is processed. The system utilizes SYBR green dye to detect the amplification product with the naked eye. The efficiency of the microfluidic chip was tested with human plasma samples spiked with HCV virions, and the limit of detection observed was 500 virions/mL within 45 min. The entire virus detection process was executed inside a uniquely designed, inexpensive, disposable, and self-driven microfluidic chip with high sensitivity and specificity.

## 1. Introduction

According to the World Health Organization (WHO), more than 354 million people worldwide are infected with the Hepatitis C virus (HCV), of which 70 million are chronically infected. Each year, an estimated 1 million people die from this disease [1,2]. According to the WHO, in the year 2019, 58 million people were reported with HCV infection worldwide [1]. During the same time, in the United States, 4136 were reported with acute infection; however, the estimated number of cases was 57,500 [3]. The Centers for Disease Control and Prevention (CDC) estimates that, in the United States, about 50% of the infected individuals are unaware of their infection [4].

HCV is primarily transmitted by parenteral drug administration, blood/plasma transfusion, the reuse of medical equipment, sexual practices resulting in blood transfer, and sharing the same needle while injecting drugs [4,5,6]. Patients are diagnosed with HCV after they exhibit symptoms arising from the chronic infection. However, patients are often asymptomatic or exhibit mild symptoms [7,8]. Currently, a combination of interferon-free and direct-acting antiviral therapy (DAA) has increased the cure rate to over 95% with fewer side effects [9,10,11]. Despite current therapies, only 20% of the population is diagnosed with the disease, and only 7% have received treatment within developed countries [12]. In developing or low-income countries, where 78–80% of the worldwide cases reside, less than 1% receive a diagnosis and treatment [13,14,15,16]. The WHO aims to reduce the HCV infection rate by 90% and the mortality rate by 65% by 2030, compared with a 2015 baseline [3]. This aim can be achieved with increased education, testing, and treatment efforts.

Early detection of the viral infection is key for a rapid and successful therapeutic outcome [17,18]. According to the CDC guidelines, HCV testing should start with an antibody assay, followed by a nucleic acid test (NAT) for RNA detection, to confirm the initial result [19]. Reverse transcriptase loop-mediated isothermal amplification (RT-LAMP), transcription-mediated amplification (TMA), and reverse transcriptase–quantitative polymerase chain reaction (RT-qPCR) are the preferred nucleic acids tests (NAT) utilized [20,21,22,23,24,25,26]. While immunoassays are cost-effective and easier to conduct than NATs, they are less sensitive and lack the ability to detect early infection [19,27]. By contrast, a NAT-based HCV RNA test detects the infection within 1–2 weeks [28]. Although RT-qPCR remains the gold standard for HCV detection from patient samples, it is time-consuming, requires trained personnel and expensive equipment, and has a long turn-around time [15]. Additionally, RNA needs to be purified before performing RT-qPCR amplification, as impurities and possible sample degradations might increase the rate of false-negative results. Thus, reliable early detection of the infection is limited for people living in low-to-middle-income and underdeveloped areas with scarce access to well-equipped medical facilities.

In recent years, HCV point-of-care (POC) testing has improved the screening and management of the disease [5,6,7]. However, these tests are expensive, provide less sensitivity, or might have a long turn-around time for the results. The most-used POC-based test is OraQuick. OraQuick is an FDA-approved rapid antibody test for HCV, but it is fairly slow (20 to 40 min for results) and expensive (USD 500 for 25 tests) [29]. In addition, a trained phlebotomist is required for blood collection [30]. Other ELISA-based POC assays include Elecsys Anti-HCV II assay, InTec assay, and Well oral anti-HCV. These assays provide inconsistent results since they are dependent on the level of anti-HCV antibodies in the patient [31,32]. NAT-based POC HCV tests include the Roche COBAS Taqman HCV test, the Hologic Aptima HCV Quant Dx assay, the Cepheid GeneXpert (not FDA-approved in the US), the Veris (Beckman), the Realtime HCV (Abbott), and the Gendrive (WHO pre-qualified). However, expensive equipment is needed to process the samples, and it can take anywhere from 90 min to 5–6 h to obtain the results [33,34,35,36]. Overall, the available HCV RNA tests are not well-suited for POC settings and community health centers. A low-cost and rapid NAT-based HCV test has yet to be developed. The nucleic acid isolation and the purification for RT-PCR require several manual steps that make this assay impractical for POC settings. An alternate method to RT-PCR is RT-LAMP, which provides an easy, scalable, low-cost, and accurate detection method where amplification can be achieved at a constant temperature with 4–6 primers [37,38,39,40,41,42,43,44,45,46,47,48]. RT-LAMP eradicates the requirement of sophisticated equipment (i.e., thermocycler), and the viral nucleic acid can be amplified with minimal purification. Previously, RT-LAMP-based POC devices have been developed; however, these devices require complicated chip assembly, manhandled processing steps, air-drying steps, and the requirement of smartphones to interpret the result [49,50,51]. Appendix A illustrates the comparison of the existing HCV detection methods in terms of target used, detection time, and limit of detection (LOD), along with the limitations.

Herein, we report a RT-LAMP-based fully automated sample-in–answer-out molecular diagnostic set-up for rapid HCV detection. Our device consists of a compact microfluidic chip that enables nucleic acid isolation, purification, amplification, and colorimetric detection of the amplified product. We utilize SYBR green 1 dye, which changes from orange to green in the presence of double-stranded DNA, resulting in easy analysis without fluorescent imaging. To test our chip, we utilized plasma spiked with HCV. The RT-LAMP-based microfluidic chip exhibited a LOD of 500 virion copies/mL within 45 min. The device is cost-effective, user-friendly, portable, and provides a visual confirmation while utilizing a small amount of sample and few reagents in a time-efficient manner. This device is, therefore, ideal for application at POC and in underdeveloped countries. Figure 1 summarizes the experimental workflow and the schematics of the microfluidic chip utilized in the assays.

## 2. Materials and Methods

### 2.1. HCV RT-LAMP Primer Design and Testing

The 5’ untranslated region (5’ UTR) of the JFH-1 isolate RNA genomic sequence (GenBank: AB047639.1) [52] [M1] was used as a target. The RT-LAMP primers were designed with the freely available online software Primer Explorer V5 from Eiken Chemical Co. Ltd (Tokyo, Japan). A BLAST search was carried out in the GenBank nucleotide of the primers. The conserved target DNA sequence was synthesized (Integrated DNA Technologies) for the initial sensitivity and specificity testing of the designed primer set. The LavaLAMP RNA Master Mix kit from Lucigen (Middleton, WI, USA) was used for all the RT-LAMP reactions (both benchtop and on-chip testing). The LAMP reactions set-up included Master Mix (12.5 μL), green fluorescent dye (1 μL), HCV RT-LAMP primers (2.5 μL), HCV, ZIKA, SARS-CoV-2, HIV or RNase/DNase-free water (1 μL), and RNase/DNase-free water (8 μL). The AriaMx Real-Time PCR system (Agilent) was utilized to maintain 70 °C for 40 min for the isothermal amplification. The LAMP-amplified products were further confirmed with 1.5% agarose gel electrophoresis. The gel was subjected to 90 volts for 90 min.

### 2.2. Analysis of the HCV-Spiked Samples by RT-LAMP

The HCV cultured samples were obtained from the University of Miami. The HCV strain was propagated in Huh 7.5.1 human hepatoma cell line. The virus titer was quantified by qPCR. Human blood samples from deidentified healthy individuals were obtained from Continental Services Group, Inc. (Fort Lauderdale, FL, USA). Under the institutional review board (IRB)-approved protocol, the plasma was separated from the blood by centrifugation for 15 min at 2000× *g*. The plasma was spiked with HCV varying from 2.8 × 10^7^ to 28 virion copies/mL. The viral RNA was isolated from the plasma samples, utilizing the Dynabeads SILANE viral Nucleic Acid (NA) kit (Invitrogen). A total of 100 μL of plasma sample was used for the RNA isolation, following the manufacturer’s instruction. The RNA was then eluted in 50 μL of elution buffer and the RT-LAMP assay was carried out in a thermocycler, as described above. Fluorescence data was collected for the endpoint analysis. In addition, the amplification products of the LAMP reaction were analyzed by 1.5% gel electrophoresis. A total of 1 μL of SYBR green I nucleic A dye (Invitrogen) was added to the reaction solution after the isothermal amplification for the colorimetric visualization.

### 2.3. Microfluidic Chip Design

Microfluidic chips were fabricated from three-layered poly(methyl methacrylate) (PMMA) sheets. The layers of the chip were attached together using double-sided adhesive (DSA) tape. Appendix A show the thickness and dimension of each layer and the chambers of the microfluidic chip. Figure 2a,b show the complete chip (top and vertical view). The chip layout was designed in AutoCAD, and a CO_2_ laser cutter was utilized to obtain the chambers, as previously reported [22,40,52,53,54]. Each microfluidic chip consists of 4 independent diamond-shaped aqueous chambers: one sample inlet chamber, two washing buffer chambers, and one reaction chamber separated by three elliptical-shaped valving chambers containing mineral oil (viscosity- 15 cSt). An unconnected oval-shaped sensor chamber adjacent to the reaction chamber is imprinted for sensor attachment. The assembled microfluidic chip was subjected to UV for 30 min. After 30 min, all the inlets were blocked with scotch tape. The scotch tape was removed from the microfluidic chip for the filling of reagents.

### 2.4. Diagnostic Platform Set-Up

The automated diagnostic platform designed and optimized by our lab is depicted in Figure 2c [55]. The diagnostic platform moves the magnetic beads automatically. The bead’s movement is directed by two small magnets (5 mm-diameter neodymium) fitted in a 3-D-printed inclusion. This 3-D-printed inclusion can move bidirectionally, and its movement is coordinated by a stepper motor. The stepper motor is guided by a printed circuit board (PCB) Arduino. The movement of the magnetic beads from one chamber to another and the incubation time were controlled by a G-code scripted in Python. We utilized an Arduino temperature controller to maintain the 70 °C temperature required for on-chip amplification [22]. The sensor is placed in the sensor chamber.

### 2.5. On-Chip Detection of HCV from Human Plasma

The microfluidic chip was loaded with reagents and buffers, as described in Appendix A. The inlet chamber (a in Figure 1) contained the lysis/binding buffer + proteinase K + magnetic beads + isopropanol. Washing buffers I and II were added to the washing chambers (b and c in Figure 1) and their viscosity was increased by adding RNase/DNase-free water in a 1:1 ratio. The reaction chamber and sensor chamber (d and e in Figure 1) solution contained LavaLAMP RNA Master Mix, HCV-specific primers, and elution buffer. The MgSO_4_ concentration of LavaLAMP RNA Master Mix was increased from 5 mM to 9.8 mM for on-chip testing. A total of 100 μL of plasma sample was loaded in the inlet chamber of the pre-filled chip and placed on the diagnostic set-up. Next, the heater and magnetic actuation were started concurrently, and the automated set-up was incubated for 45 min. SYBR green 1 dye was added to the reaction chamber after the isothermal incubation and mixed using the oscillatory movement of the magnetic beads to detect the reaction amplification products. All the on-chip experiments were repeated at least three times.

### 2.6. Results and Discussion

HCV is comprised of seven distinct genotypes, which can differ by as much as 30% at the nucleotide level [56,57,58]. Though sequence variations are evenly distributed throughout the genome, the 5’ UTR is conserved [59,60,61]. The HCV RT-LAMP primers we designed (Table 1) are mapped on the highly conserved 5’ UTR region. The designed primer set was BLAST, against all the genomes present in the National Center of Biotechnology Information (NCBI). The results showed high diversity against other genomes and identify to the HCV genome 5’ UTR region.

We performed an initial RT-LAMP assay to test the sensitivity and specificity of the primer set we designed (Figure 3A), utilizing a synthetic target HCV DNA sequence with different target concentrations (10^9^ to 0 copies per reaction). The lowest LOD observed was 5 HCV copies/reaction within 40 min. A no-template control (NTC) confirmed the absence of primer dimer formation. The R^2^ value (0.967) of the time to amplification relative to the concentration of the target substrate indicates that the amplification time of the target is inversely proportional to the target concentration in the reaction. These results were confirmed by analyzing the LAMP products by agarose gel electrophoresis. Amplification products can be observed in the reactions that amplify the target HCV sequences, forming long sharp bands (Figure 3B). The HCV primer set was tested by amplifying the cDNA obtained from the RNA genome of ZIKA virus, [22] SARS-CoV-2, [62] and HIV [63] (Figure 4 and Appendix A). The HCV primer sets did not amplify viral sequences from these viruses, confirming the specificity of the RT-LAMP primers.

As a proof-of-principle, human plasma was spiked with a cultured HCV strain to replicate clinical samples. HCV preparations were added to the plasma samples with 10-fold dilutions, obtaining final concentrations ranging between 2.8 × 10^7^ and 28 virions/mL. The viral RNA was extracted from the spiked plasma samples utilizing magnetic Dynabeads. The RT-LAMP of the samples yielded a LOD of 28 virions/mL within 60 min (Figure 5a). From this assay, a fitted linear trendline was observed with R^2^ = 0.865, indicating the high efficiency of the RNA extraction/RT-LAMP analysis with limited amounts of virus. These results validate the efficiency of the magnetic beads’ RNA extraction and the fact that the RNA maintains its integrity throughout the purification steps. Appendix A represents the gel results. A colorimetric test was performed utilizing SYBR green I dye, which is initially orange but turns to green in the presence of DNA, to enable a fast and simpler readout [64]. The SYBR green I was added to the reaction tubes after the isothermal incubation and confirmed the LOD for the assay for the samples containing 28 virion copies/mL, which showed an attenuated shift to the green coloration when compared with the control reactions that retained the original orange color (Figure 5b).

In recent years, molecular detection microfluidic platforms have transformed the healthcare landscape, as they offer the rapid detection of viruses and other pathogens [65,66,67,68,69]. Herein, we have designed a microfluidic platform that incorporates different steps that are usually performed by trained personnel with sophisticated lab settings on a single platform. The microchip is designed with distinct shapes of chambers so that the solutions can be retained during the entire execution process. The diamond-shape chambers contain different buffers performing different tasks for optimal RNA purification. The inlet and reaction chambers are dual-purpose chambers, making the plasma processing steps less complex. These diamond chambers are separated by elliptical chambers which contain mineral oil. The oil reduces the interfacial energy barrier that facilitates the smooth passage of the beads from one buffer to another. The geometry of the chambers is designed in such a way that the magnetic beads align with magnets situated on the diagnostic platform. This provides an approachable magnetic field to the beads for flexible RNA isolation processing. Using a 750 µm PMMA sheet at the bottom also contributed to the maximum magnetic force, as the distance between the magnetic beads and the magnet was minimized. The magnetic beads used for the isolation process are composed of cross-linked polystyrene and magnetic material. Silica-like surface chemistry offers excellent binding of RNA, and the magnetic property of the beads aids with fast mobility under the magnetic field. Furthermore, given the beads’ low sediment rate, no adhesion was observed at the bottom. The incubation of the beads in each chamber is facilitated by oscillatory movement directed by a stepper motor on the platform, which is controlled by a G-code. Appendix A represents the beads’ incubation time in each chamber, directed by the automated magnets located on the platform. For on-chip isothermal temperature, an automated Arduino-based temperature controller is used, which consists of a K-type thermocouple sensor and an ultrathin nano-carbon flexible heater. A K-type thermocouple sensor reads the real-time temperature. The heater heats the reagents in the sensor chamber and reaction chamber. To maintain the set isothermal temperature, the sensor reads the temperature every 2 s. Once the temperature is higher than the set temperature, the sensor turns “off” the heater, and if the temperature is lower than the set temperature, it turns “on” the heater. To avoid direct contact between the sensor and the reaction chamber solution, the sensor was inserted into the sensor chamber. The sensor chamber contains the same reagents as the reaction chamber, and one surface heater was attached on the top of both chambers. RNA degradation could lead to false-negative results in an RNA-based detection device. To address this potential problem, we exposed the RT-LAMP microfluidic chip to UV for 30 min, which eliminated traces of RNases from solid surfaces [70,71].

On the microfluidic chip, the key role of the inlet chamber is to lyse the viral particles and to allow binding of the viral RNA to the magnetic beads in the presence of lysis and binding buffer. Once the RNA adheres to the surface of the magnetic beads, they are directed toward the reaction chamber under the influence of a magnetic field. The two washing chambers contain two different washing buffers to ensure that reagents and other potential inhibitory compounds are not transferred from the inlet chamber to the reaction chamber. Once the beads reach the reaction chamber, which is already maintained at 70 °C with the help of the sensor chamber and surface heater, beads elute the RNA and move back to the washing chamber. The RT-LAMP HCV primers amplify the HCV RNA if present, and change the color of the chamber to green; if HCV RNA is not present initially, the color of the dye will remain unchanged.

For the on-chip assay, plasma samples were spiked with different amounts of HCV concentration (from 2.8 × 10^6^ to 282 virions/mL). A no-template control assay was carried with non-spiked plasma samples. Since PMMA adsorbs polar molecules, such as DNA and polymerase enzymes that inhibit amplification [72], the addition of ions such as Mg^++^, Na^+^, and K^+^ enhances the enzyme activity and provides DNA stability [73,74]. To enhance the on-chip amplification efficiency of the target, the MgSO_4_ concentration of the Master Mix was increased from 5 mM to 9.8 mM in the reaction chamber. Figure 6a shows the color comparison of the reaction chambers of the microfluidic chip subjected to a non-spiked plasma sample (orange color in the reaction chamber) and HCV-spiked sample (green color reaction chamber) after 45 min. Figure 6b demonstrated the on-chip sensitivity of the set-up: 500 virions/mL within 45 min. At the same time, no color change was observed in the negative control (non-spiked plasma) sample and 280 virions/mL. These results validate that the RT-LAMP-based microfluidic chip we developed can efficiently isolate and amplify HCV RNA. The molecular diagnostic set-up here described provides qualitative colorimetric results from plasma samples spiked with HCV. The data presented demonstrate that this compact microfluidic device could be utilized as a “sample-in–answer-out” system for HCV screening in low-to-middle-income areas.

POC microfluidic platforms offer comparable results to gold-standard methods using a small quantity of reagents and the sample can be tested outside the laboratory setting. A commercial device derived from the one described here, if utilized broadly, can play an essential role in managing HCV infection worldwide, thereby controlling the spread and enabling the timely treatment and monitoring of the disease. The 3-D-printed platform used in the research consists of a microprocessor controlled by Arduino, a step-up motor, and a circuit for the power supply. The molecular diagnostic platform is fully programmed and reusable for repetitive testing. It can also be operated using batteries to direct the movement of magnetic beads present in the chambers of the disposable microfluidic chip. The operational cost of the diagnostic set-up would be minimal, as it uses relatively inexpensive and sustainable equipment (roughly USD 50) for sample processing and disease detection. This microfluidic chip also offers shorter times for a reliable diagnosis of HCV infection. The disposable microfluidic chip we describe is affordable (roughly USD 2) and can be utilized in POC settings, as the assay is performed in an automated fashion. With this molecular diagnostic set-up, the user can run multiple tests since attention is required only initially and at the end of the colorimetric analysis. Appendix A contains the full list of materials and costs for the molecular diagnostic microfluidic chip set-up and fabrication. In this study, we developed a low-cost RNA-based POC molecular diagnostic set-up for HCV with a colorimetric result readout. The developed diagnostic method will enable timely HCV screening and prevention in high-risk populations. It is an accurate method that can be implemented in low-income areas, making it accessible to people.

## 3. Conclusions

A lack of overt symptoms in HCV patients often results in a lack of diagnosis, which subsequently leads to increased morbidity and mortality. Therefore, it is important to implement regular screening in high-risk populations. Molecular testing is considered the most accurate test for diagnosing HCV infection. Our research describes a single-step procedure for the molecular detection of HCV RNA from a patient’s samples with clinical applicability. The microfluidic chip utilizes an automated system for the hands-off processing of plasma samples. The testing of human plasma samples spiked with HCV particles showed that this set-up offers a sensitivity of 500 viral copies/mL and high specificity, without the need for trained technicians, expensive equipment, or facilities. The hands-free microfluidic chip used for the testing is easy to assemble, low-cost, and provides a practical approach for large-scale testing outside the laboratory. The operating procedure of the chip is straightforward; once the plasma samples are introduced to the inlet chamber, the automated system will self-operate. The cost reduction, need for less equipment, and short time (45 min) required for viral RNA detection offered by this approach would substantially reduce the financial and operational burden of large-scale testing in low-to-middle-income countries. Overall, the HCV diagnostic test approach we propose can be clinically implemented for routine HCV screening and can help in the timely execution of preventive and therapeutic steps to limit HCV spread.

## Figures and Tables

**Figure 1 biosensors-12-00298-f001:**
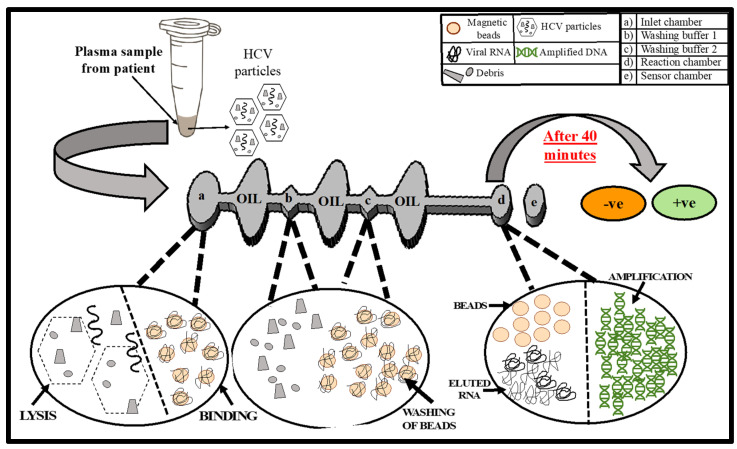
The complete workflow of the developed HCV diagnostic method. The layout represents the compact microfluidic chip, consisting of different chambers along with their specific functions. Accurate and visible colorimetric analysis can be obtained within 45 min. Green for HCV +ve and orange for HCV −ve.

**Figure 2 biosensors-12-00298-f002:**
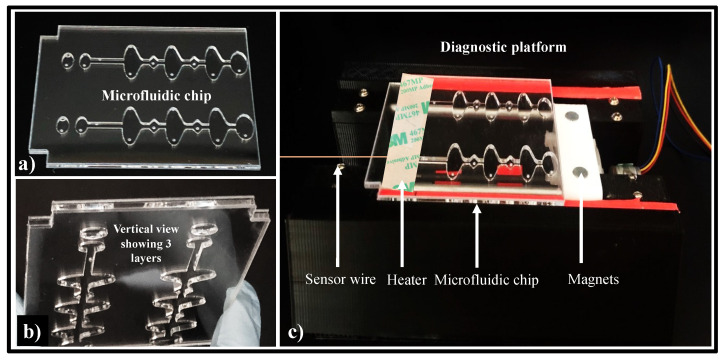
(**a**) Fully assembled compact microfluidic chip. (**b**) Vertical view of the chip, showing 3 different layers—top, middle, and bottom layer. (**c**) Representation of the automated molecular diagnostic set-up.

**Figure 3 biosensors-12-00298-f003:**
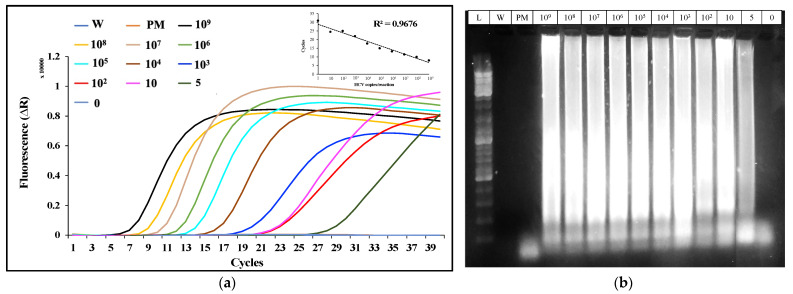
(**a**)**:** Amplification plot for the HCV primers’ sensitivity and specificity assay with different target concentrations per reaction. All the reactions carrying the HCV target showed signs of amplification, and no amplification was observed in the primer + Mastermix (PM) reaction and the reaction containing no HCV target. Baseline-subtracted amplification points (top right) of the HCV target represent the change in the fluorescent intensity over time (R^2^ = 0.9676). (**b**)**:** The 1.5% gel electrophoresis results stained with Bromophenol blue dye (lane L contains a 1 kbp-size DNA ladder). Sharp bands in the wells holding the LAMP amplification product of HCV with 10^9^ to 5 copies/reactions clearly show the designed primers’ specificity and provide the sensitivity of up to 5 copies/reactions.

**Figure 4 biosensors-12-00298-f004:**
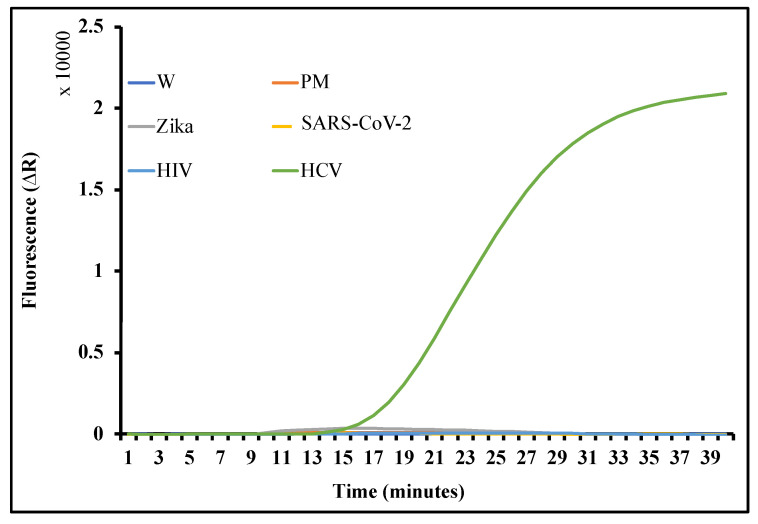
Specificity graph of designed RT-LAMP primers. An increase in fluorescent value is observed only in the reaction carrying HCV target and negative controls—ZIKA, COVID-19, and HIV showed no sign of amplification.

**Figure 5 biosensors-12-00298-f005:**
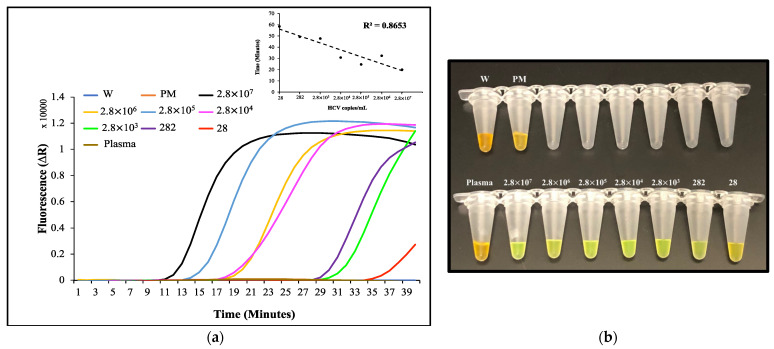
(**a**): Real-time detection of the human blood plasma spiked with HCV virions with various copies/mL. The curves showed the abrupt rise in signal with the reaction-carrying HCV targets (2.8 × 10^7^ to 28 HCV copies/mL) isolated from plasma. No signal was observed in negative control reactions, including non-spiked plasma and primer + Mastermix (PM). Baseline-subtracted amplification points (top right) of the plasma-spiked HCV target/mL, representing a change of fluorescent intensity over time (R^2^ = 0.8653). (**b**): Off-chip RT-LAMP colorimetric detection test results of the plasma spiked with HCV virions. Color change from orange to green was observed due to the amplification in the reactions (2.8 × 10^7^ to 282, and slightly in 28 HCV virion copies/mL) using SYBR green I dye.

**Figure 6 biosensors-12-00298-f006:**
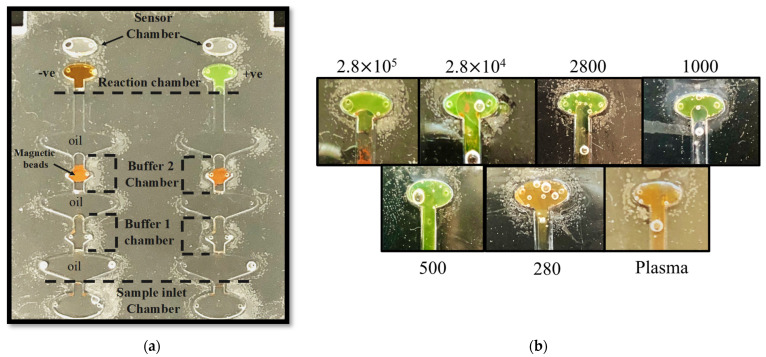
(**a**): Photo of the RT-LAMP HCV microfluidic chip after 45 min of sample processing and isothermal incubation with the developed magnetic actuation platform. The orange color in the reaction chamber represents that the plasma sample was HCV-negative, and the green color signifies the plasma sample was HCV-positive; (**b**): image of the reaction chamber after the fully automated chip run of the HCV-spiked (virion copies/mL) and non-spiked plasma samples. The LOD of the set-up is 500 virions/mL within 45 min.

**Table 1 biosensors-12-00298-t001:** List of HCV RT-LAMP primer sequences alongside labeled sequences, concentration, Tm, and delta G values.

Sequence Name	Sequence 5′ to 3′	Concentration (μM)	T_m_	5′d ΔG	3′d ΔG
**F3**	AAACCCACTCTATGCCCG	4	58.3	−5.11	−7.53
**B3**	TACTCCGCCAACGATCTG	4	57.9	−4.43	−4.25
**FIP**	GCCCTATCAGGCAGTACCACAAGCAAGACTGCTAGCCGAG	32			
**BIP**	GCACCATGAGCACAAATCCTGGGAACTTAACGTCTTCTGG	32			
**LF**	GCCTTTCGCAACCCAACGCTA	16	65.6	−5.35	−5.65
**LB**	AAACCAAAAGAAACACCAACCGTCG	16	64.6	−4.67	−6.96

## Data Availability

All data needed to evaluate the conclusions in the paper are present in the paper and/or the Appendix A. Additional data related to this paper may be requested from the authors.

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
