# Peer review of "RT-LAMP-Based Molecular Diagnostic Set-Up for Rapid Hepatitis C Virus Testing"

_biosensors, 2022, doi:10.3390/bios12050298_

Round 1

Reviewer 1 Report

This study presents an automated RT-LAMP diagnostic set-up for HCV detection in plasma samples. This is an interesting design for POC diagnosis as it is relatively inexpensive given the achieved high sensitivity and selectivity with relatively short detection time. I have a few minor comments to the authors to address in the revised manuscript:

1- It would be good to have a summary table that compares this design with other sensor technologies showing detection limit, cost, detection time, power consumption etc in each platform.  

2- What is the throughput? is it a reusable platform? I think it would be useful to discuss them in the manuscript.

3-  I recommend to add more description about the design of different parts such as the magnetic actuation platform. This could be added to the supporting information.

Reviewer 2 Report

In this study, Sharma and colleagues developed an RT-LAMP based microfluidic chip for Hepatitis C virus testing. Although the study seems rational, they exist some challenges that the authors should clarify before further considering their article.

First is that the literature does not truly reflect the article content and the history of research development on the Hepatitis C virus testing. In the introduction section, the authors mention that “The most used POC-based test is OraQuick. OraQuick is an FDA-approved rapid antibody test for HCV, but it is fairly slow (20 to 40 minutes for results) and expensive ($500 for 25 tests).” Nevertheless, the test developed by the authors has 45 min operation time which is more than the test mentioned here. A proper judgment of the literature and products is essential.

The second is that the articles for rapid detection of Hepatitis C are mature. Many articles have been developed, from early 2000 up to now, for rapid detection of Hepatitis C. Authors should provide a table and compare the study they developed with other articles to make sure they correctly reflect the value of their article compared to other available studies and products.

The third is the use of double adhesive tape for the current study. Double adhesive tapes are not ideal for such experiments if they are not biocompatible. From Figure 6 of the article, it is evident that the bonding is not proper, and near the channel sites, there are some points where the channels are not properly bonded.

Also, better western blot figures must be provided in the article. According to what was mentioned, I do not recommend the article with the current format for publication in the journal of Biosensors.

Round 2

Reviewer 2 Report

The authors address the issues raised by the reviewer.